# Empowering Future HR Professionals: A Design-Based Research Approach to Project-Based Learning in Work and Organizational Psychology

Sabrina Krys * and Mirjam Braßler *

Work and Organizational Psychology, Institute of Psychology, Kiel University, 24118 Kiel, Germany
* Correspondence: krys@psychologie.uni-kiel.de (S.K.); brassler@psychologie.uni-kiel.de (M.B.)

**Abstract**

This study reports on a Design-Based Research (DBR) project that implemented Project-Based Learning (PjBL) in an undergraduate psychology course on Human Resource Development (HRD). The purpose was to move beyond lecture-based instruction and explore how open pedagogy can create authentic, student-centered learning experiences that bridge theory and practice. Over two course iterations ($n = 31$), students co-designed, implemented, and evaluated HRD interventions for their peers, supported by peer and instructor feedback and complemented by a co-created open-book exam. Quantitative pre- and post-tests revealed significant improvements in students' knowledge of HRD methods, learning theories, and application competencies, as well as enhanced confidence in their professional qualifications. Students valued the openness of the design, its practical orientation, and the error-friendly learning environment, though challenges emerged regarding workload, communication, and intrinsic motivation. Educators reported a transformation of their role from knowledge transmitter to facilitator and co-learner, while also identifying opportunities to use AI for generating authentic case tasks. The findings suggest that PjBL, combined with open pedagogy, fosters self-directed learning, transparency, and collaboration, thereby contributing to cultural change in higher education toward openness, participation, and innovation.

**Keywords:** project-based learning; open pedagogy; higher education development; human resource development; design-based research; student-centered learning; psychology education; openness; academic culture transformation

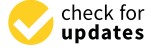

## 1. Introduction

*"It is important to me to gain a comprehensive insight into human resource development. I would highly value initial practical exercises to move beyond 'dry' theory and establish a real-world connection."* (student quote)

This quote reflects a frequently voiced student desire for authentic, practice-oriented learning that bridges academic theory and real-world application in higher education. In line with this demand, contemporary learning principles and theories highlight the importance of active engagement, social interaction and collaboration, feedback, and knowledge transfer for meaningful and sustainable learning outcomes (Biggs & Tang, 2011; Chi & Wylie, 2014; Sotto, 2021). Project-Based Learning (PjBL) is a pedagogical approach that integrates these principles. It shifts from traditional, lecture-based instruction toward learning

formats in which students work on complex, real-world tasks and inquiries, often culminating in a tangible product or presentation (Bell, 2010; Blumenfeld et al., 1991; Boss & Larmer, 2018; Guo et al., 2020; Markham et al., 2003; Thomas, 2000). Rooted in "learning by doing" (Dewey, 1938) and social constructivism (Vygotsky, 1978), PjBL fosters student-centered learning that promotes deep understanding and personal relevance by enabling students to investigate and respond to authentic questions, problems, or challenges over an extended period, for example, one semester (Jaleniauskiene & Venckiene, 2025; Larmer, 2014).

Previous research has demonstrated the effectiveness of PjBL in promoting student-centered learning. It places students at the center of their educational experience and transforms educators into facilitators—"guide on the side" (Wright, 2011, p. 93)—who support rather than control the learning process. This shift encourages active engagement, personal relevance and a stronger sense of responsibility for learning outcomes.

Importantly, PjBL also fosters self-regulated and self-directed learning as students are required to define project goals, manage timelines, monitor progress, and reflect on their learning strategies (Clausen, 2023; Lai, 2021; Wu, 2024). By involving students in practical tasks such as research, case studies, and collaborative problem solving, PjBL increases their motivation and intrinsic engagement, making learning feel personally relevant (van der Walt & Bosch, 2025). This collaborative environment also sharpens important social and communication skills, as students exchange ideas and perspectives (Guo et al., 2020). It promotes peer learning and exposes the students to different points of view. Ultimately, these elements contribute to developing a deeper and more comprehensive understanding of the subject matter (Sulong et al., 2023).

Research across various disciplines, including higher education psychology, demonstrated that PjBL yields significant positive effects on several student outcomes. These include deeper subject-related knowledge, a variety of skills such as problem-solving and creativity, improved learning and academic achievement, and increased satisfaction and engagement (e.g., Guo et al., 2020; Smolyaninova et al., 2021; Sulong et al., 2023; Zhang & Ma, 2023). Furthermore, PjBL fosters motivational aspects, such as competence beliefs, intrinsic motivation, and future interest (Wijnia et al., 2024). These findings underscore the effectiveness of PjBL as an instructional method in psychology education, fostering active participation and exploration.

In response to these findings—and in alignment with current calls for cultural transformation in academia—we redesigned our courses on Human Resource Development (HRD) to adopt a more participatory, student-driven model of teaching and learning. This article presents a systematic account of this transformation, guided by a Design-Based Research (DBR) approach (McKenney & Reeves, 2018). DBR enabled us to iteratively analyze, implement, evaluate, and reflect upon the new course format in its authentic context.

The structure of the article mirrors this process: it begins with a practitioner's reflection on the previous course design (Status quo Ante), followed by the implementation of a redesigned PjBL intervention, a preliminary and exploratory empirical evaluation of its impact, and a second reflective phase to derive implications for future design cycles. The course described here, also known as personnel, employee or–more generally–people development, is part of the required curriculum in work and organizational psychology in Bachelor's degree programs accredited by the (German Psychological Society [DGPs], 2023). Its redesign was guided by the principle of constructive alignment (Biggs & Tang, 2011): ensuring coherence between learning outcomes, teaching and learning activities, and assessment methods.

The purpose of this article is threefold. Firstly, it aims to demonstrate how PjBL, informed by open pedagogy, can be meaningfully integrated into higher education psychology courses to foster participatory and authentic learning. Secondly, the article seeks to

critically reflect upon implementation challenges and to derive practical design strategies that support sustainable and inclusive teaching practices. Thirdly, the article contributes to the evolving field of open and student-centered educational design through a DBR-informed approach. More precisely, it presents promising practices and preliminary data in the context of open education, aiming to promote knowledge sharing, embrace transparency and collaboration, and contribute to the cultural transformation of academic practices toward greater equity, sustainability, and innovation.

The guiding research questions are: (1) How can PjBL, informed by principles of open pedagogy, be designed and implemented in psychology-based higher education contexts such as HRD to foster openness and student participation? (2) What advantages, challenges, and cultural implications arise from this approach for both students and educators, particularly with regard to transparency, collaboration, and the transformation of academic practices?

## 2. Open Pedagogy as a Framework for Educational Innovation

Open pedagogy has gained increasing attention as an approach that repositions students from passive recipients of knowledge to active participants in learning processes (Cronin, 2017; DeRosa & Robison, 2017). Rather than focusing only on content transmission, open pedagogy emphasizes shared responsibility, learner involvement, and the creation of knowledge that connects to broader social and professional contexts (Bali et al., 2020).

To ensure conceptual clarity, we summarize in Table 1 the principles of open pedagogy as derived from the literature we draw on (e.g., Bali et al., 2020; Bovill, 2020; Cronin, 2017). These principles provide the framework for our course design and the basis for the reflections presented later in this manuscript.

**Table 1.** Principles of open pedagogy pursued in the present study.

| Principle | Definition | References |
|---|---|---|
| Authentic knowledge practices | Creation, sharing, and application of knowledge in real-world contexts with authentic audiences. | (Bali et al., 2020; Cronin, 2017; DeRosa & Robison, 2017) |
| Co-creation (i.e., participatory design) | Students and educators act as partners in curriculum, teaching, or assessment design. | (Bovill, 2020) |
| Collaboration (i.e., participatory or co-learning) | Learning as a social and participatory process through shared inquiry and knowledge-building. | (Bali et al., 2020; DeRosa & Robison, 2017) |
| Democratization and transparency | Teaching processes are open, negotiated, and co-constructed with students. | (Bovill, 2020) |
| Formative learning cycles | Iterative feedback, reflection, and revision emphasize learning as ongoing. | (Bali et al., 2020; DeRosa & Robison, 2017) |
| Inclusivity | Open pedagogy values equity, diversity, and multiple perspectives. | (Bali et al., 2020; Bovill, 2020) |
| Learner agency/autonomy | Students make meaningful choices and take responsibility for their own learning. | (Cronin, 2017; DeRosa & Robison, 2017) |

Motivational theories help to substantiate these principles by explaining why they are pedagogically powerful. For example, self-determination theory (Ryan & Deci, 2000) suggests that open pedagogy's emphasis on learner agency, collaboration, and authentic knowledge practices supports students' basic psychological needs for autonomy, com-

petence, and relatedness. Expectancy-value theory (Eccles & Wigfield, 2002; Wigfield & Cambria, 2010) likewise highlights how transparency and authenticity increase students' expectancy for success and the perceived value of their tasks. Achievement goal theory (Elliot & McGregor, 2001; Urdan & Kaplan, 2020) further explains how open and participatory learning designs promote mastery goals rather than performance goals. Finally, flow theory (Csikszentmihalyi, 1990) shows how authentic, optimally challenging tasks that balance skills and difficulty can foster deep engagement. Together, these perspectives provide a psychological rationale for why open pedagogy fosters engagement and intrinsic motivation.

We conceptualize PjBL as an enactment of open pedagogy. Whereas open pedagogy provides the pedagogical foundation (e.g., values of co-creation, transparency, learner agency), PjBL offers the methodological vehicle through which these values are operationalized in practice and linked to motivational dynamics such as autonomy and deep engagement. The design invited students to take responsibility for their own HRD interventions, collaborate in addressing authentic challenges, and contribute to assessment through co-created exam questions. Framing the study in this way allows us to connect course-level design decisions with broader theoretical debates about how open pedagogical approaches can be implemented in higher education and the institutional tensions—such as workload, assessment requirements, and traditional hierarchies—that may accompany them.

Building on this framework of open pedagogy, the following section outlines the status quo ante of the course to illustrate why a redesign was necessary.

## 3. Status Quo Ante

To initiate the design-based research (DBR) process, we conducted an in-depth reflection on our prior teaching practices in HRD. Following Zeichner and Liston's (2013) approach to reflective teaching and the DBR framework by McKenney and Reeves (2018), this reflection served both to identify contextual challenges and to inform the development of a new, student-centered learning design. The practitioner's reflection encompassed multiple dimensions—including technical, situational, didactic–pedagogical, ethical, and critical–political perspectives—and functioned as the analytical foundation for subsequent design iteration. Our DBR cycle is visualized in Figure 1.

Previously, the HRD courses were primarily lecture-based and assessed through student-led oral presentations. The objective was for students to introduce specific HRD interventions (e.g., team development, coaching and mentoring, stress management, diversity training, self- and time management), incorporating relevant scientific evidence on their effectiveness. These presentations were the sole form of assessment. Consequently, students received instruction on conceptualizing HRD interventions across several sessions, culminating in a small-group presentation to their peers at the course's end. While this format required relatively little educator effort, it also presented several limitations from a reflective teaching perspective, which are elaborated below across different reflection dimensions.

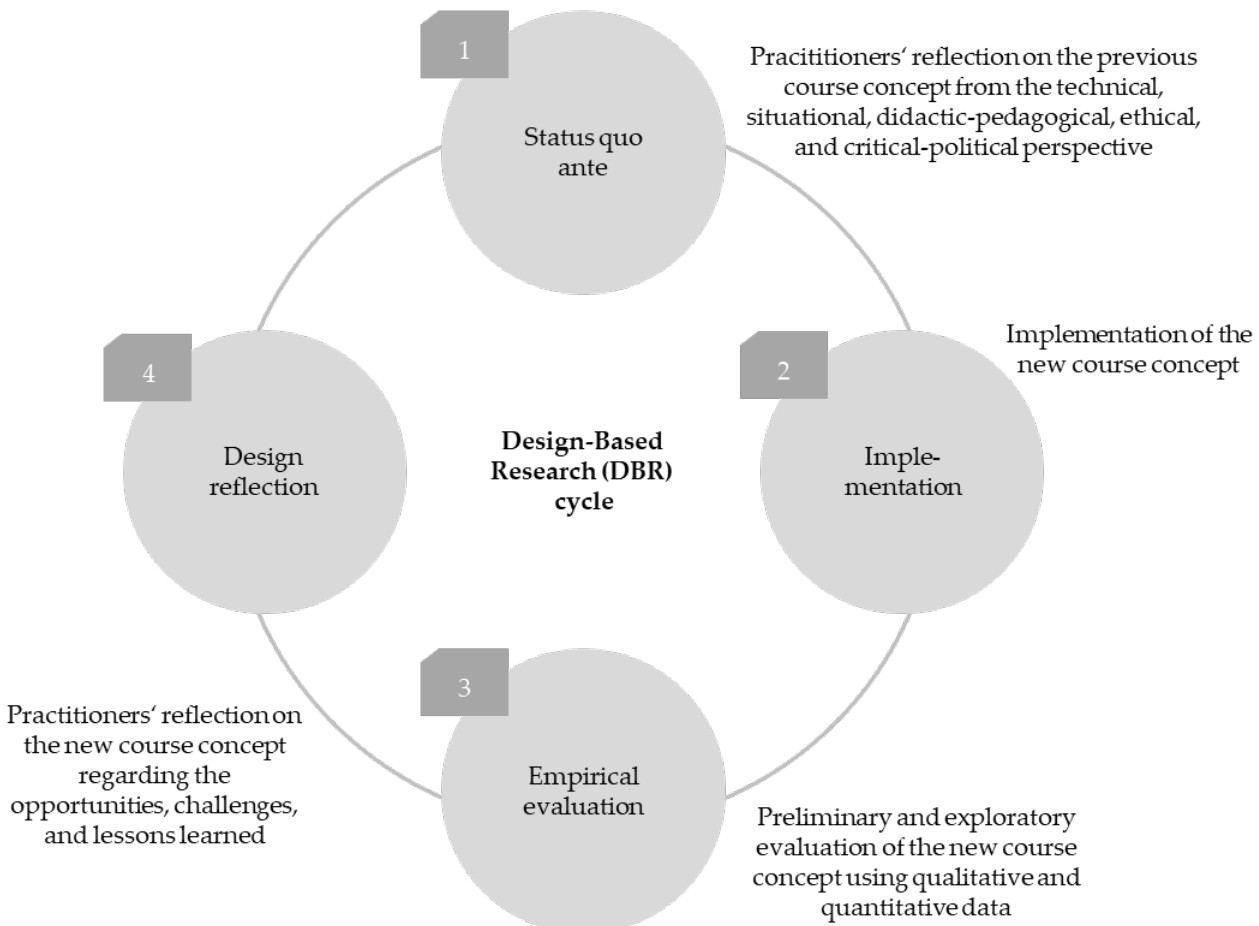

**Figure 1.** Visualization of the Design-Based Research (DBR) cycle applied in the present study (adapted from McKenney & Reeves, 2018; Zeichner & Liston, 2013).

### 3.1. Technical Perspective

From a technical perspective, the method proved ineffective in promoting deep learning (Asikainen & Gijbels, 2017): While students engaged deeply with their assigned presentation topic, their understanding of the conception process and its challenges remained superficial. Students thus acquired only theoretical knowledge without opportunities for application and reflection. The absence of direct application hindered knowledge transfer beyond memorization, resulting in potential limited long-term learning gains.

### 3.2. Situational Perspective

The situational context further constrained the learning experience. As the final grade was exclusively tied to the presentation, students' engagement in other course elements remained low. The educator's role during these sessions became largely passive, relying on students for content delivery. This dynamic also limited the educator's ability to scaffold learning effectively or to provide formative feedback.

### 3.3. Didactic–Pedagogical Perspective

From a didactic–pedagogical perspective, the course design restricted student autonomy and active learning. Upfront instructional control, combined with a lack of iterative, practice-based tasks, left little room for co-construction of knowledge or meaningful peer interaction. Despite repeated efforts to incorporate interactive elements throughout the semester, students lacked sufficient structure and opportunity to build and refine their understanding collaboratively.

*3.4. Ethical Perspective*

Ethically, the course structure discouraged experimentation and learning from mistakes. The high-stakes nature of a single graded presentation discouraged risk-taking and the acceptance of mistakes, both of which are fundamental to genuine learning and comprehension (Schellekens et al., 2021). Without formative stages or feedback loops, students were offered few opportunities to grow through error, despite the importance of error culture in education.

*3.5. Critical–Political Perspective*

Finally, from a critical–political perspective, the instructional model reflects broader systemic tendencies within academia: namely, efficiency-driven course designs that prioritize measurable outcomes over transformative learning (Giroux, 2014). The exclusive reliance on summative assessment and teacher-centered delivery mirrors traditional academic power structures, which can limit students' sense of ownership and suppress participatory culture. These institutional constraints—while convenient—stand in tension with values of openness, co-creation, and innovation in education.

## 4. Breaking New Ground

After several years of employing the previous course format, we recognized the need for a fundamental pedagogical shift—one that would align with evolving academic values and contribute to a broader cultural change in higher education. This realization marked the starting point of our DBR process, in which we set out to reimagine teaching practices by aligning them with principles of openness, participation, and authentic learning. Our overarching aim was to create a learning environment that fosters students to actively engage, think critically, and directly apply acquired knowledge. In line with experiential learning theory (Kolb & Kolb, 2005; Yardley et al., 2012), we placed particular emphasis on learner autonomy, self-direction, and the educational value of making and learning from mistakes without the constraint of punitive grading. For these reasons, we adopted a PjBL approach as the central pedagogy within our redesigned HRD course. This shift reflects a deliberate departure from transmissive models of instruction toward a co-constructive, student-centered learning culture that values openness, peer collaboration, and iterative design.

Table 2 provides a concise overview of this concept, with its individual components detailed in the subsequent sections (for a similar overview on PjBL, see Braßler & Dettmers, 2017, Table 2).

**Table 2.** Implementation of PjBL in a course on HRD with undergraduate psychology students in higher education.

| | |
|---|---|
| Project description | - Real-world scenario, fully authentic task co-created with students: designing a training program for fellow psychology students or rather future psychologists that are about to start their career, grounded in real needs and learner-driven decisions |
| Learning objectives | - Knowledge of how HRD interventions should be carried out from a scientific perspective<br>- Knowledge of the various methods that can be used in HRD interventions<br>- Knowledge of learning theories to understand how people learn<br>- Knowledge of the catalysts and barriers to the effectiveness of HRD interventions<br>- Ability to practically apply the acquired knowledge and develop solutions to real-world tasks including sharing knowledge transparently across peer groups and promoting sustainable learning practices |

**Table 2.** *Cont.*

| | |
|---|---|
| Group work and process | - Students are organized into small groups (3–4 members)<br>- They are encouraged to autonomously collaborate; a team charter can help organize group work<br>- Group work should take place weekly in the course itself and outside the course<br>- Students switch roles across the course: learners, designers, implementers, participants, and reviewers—encouraging perspective-taking and mutual understanding across roles, fostering inclusion and participatory learning structures<br>- The process follows general, broad steps of project management<br>  1. Task analysis<br>  2. Identification of solutions<br>  3. Implementation of solution |
| Instructor's role | - The educator acts as a teacher, product-oriented supervisor/instructor/client, and resource provider<br>- The task is mostly defined by the educator |
| Timeline and milestones | - Duration: one semester, approximately 3 months<br>- Key deadlines: peer-to-peer-feedback, educator feedback, implementation of the training, submission of the project report to the client, written open-book exam–all structured as iterative learning and reflection points |
| Assessment methods | - Not graded, group assessment: group work, peer evaluations, educator evaluations, implementation, project report—promoting a feedback culture centered on formative assessment and error-friendly learning<br>- Graded, individual assessment: written open-book exam with partial student participation in question design to enhance ownership and transparency |

### 4.1. Implementation of PjBL Elements

At the beginning of the course, students form small, stable groups that collaborate throughout the semester. In each session, the educator introduces new theoretical knowledge regarding the conception of an HRD intervention. These sessions are designed to be highly interactive, moving beyond traditional lectures to actively involve students in every stage. Following each input phase, students immediately apply the newly acquired knowledge by working collaboratively on their group project: conceiving, implementing, and evaluating an HRD intervention tailored to real needs. This iterative process, encompassing all steps, is meticulously documented by each group in a comprehensive project report, forming both a product and a reflection tool. This dual focus on knowledge application and reflective documentation exemplifies core principles of open pedagogy (Bali et al., 2020; Bovill, 2020; DeRosa & Robison, 2017), for example co-creation, collaboration, transparency, and formative learning cycles.

Within this framework, the educator assumes a hybrid role—acting simultaneously as teacher, facilitator, and client. Students design and implement an HRD intervention specifically for their fellow students within this course. This arrangement positions the groups as "service providers" offering tailored HR solutions. The client's directive is to create an intervention for sixth-semester Bachelor's psychology students, focusing on essential professional competencies expected of aspiring psychologists, emphasizing key professional competencies such as self-reflection, teamwork, and ambiguity tolerance.

Consequently, students actively engage in multiple roles throughout the project: learner, designer, implementer, participant, and reviewer. By rotating through these roles, students experience multiple perspectives—a process that fosters empathy, collaboration, and interdisciplinary skill development. This multi-role engagement aligns with student in-

volvement in the design process and supports cultural change toward shared responsibility in academic learning.

The weekly structure enables continuous progress and encourages consistent peer collaboration with meaningful autonomy. Educator support is provided on request, reflecting a shift toward learner agency and trust in student-led processes. Furthermore, the educator supports group cohesion and effectiveness by consistently referencing a team charter (e.g., mission statement, roles and responsibilities, conflict resolution strategies; Mathieu & Rapp, 2009), which groups establish at the onset of their collaboration and update regularly. This tool not only enhances group cohesion but also reflects the open education principle of co-learning, learner agency, and inclusivity.

### 4.2. Assesment Methods

#### 4.2.1. HRD Intervention and Project Report

The project report serves as an ungraded group assessment designed to emphasize learning through iteration rather than performance, reflecting the open pedagogy principle of formative learning cycles (Bali et al., 2020; DeRosa & Robison, 2017). Student groups retain substantial autonomy in the intervention's design, reflecting the open pedagogy principle of learner agency (Cronin, 2017; DeRosa & Robison, 2017). The educator offers only minimal prescriptive guidelines to encourage creative ownership. This report undergoes an interim friendly peer-review process (also known as peer-to-peer, student peer review or peer editing; Crowe et al., 2015) allowing students to provide and receive constructive feedback from peers. Subsequently, the educator provides additional formative feedback before final submission, acting in the dual role of facilitator and client. Ultimately, as the client, the educator receives the completed project report. This multi-stage assessment reflects a shift toward collaborative knowledge construction and participatory evaluation, empowering students to experience assessment *as learning* rather than *of learning* (Schellekens et al., 2021).

#### 4.2.2. Written Examination

To comply with curricular requirements for summative assessment, a written open-book exam (OBE; Durning et al., 2016) is administered at the course's conclusion. Unlike traditional exams, the OBE assesses deep understanding, application, and knowledge transfer rather than rote memorization. Importantly, the exam also reflects core ideas of open assessment (co-creation) by actively involving students in the design process: under guidance, they co-create a question pool from which 25% of the exam questions are selected (Papinczak et al., 2011). These questions vary in format (e.g., single-choice, free-text responses) and are aligned with different levels of Anderson and Krathwohl (2001) learning objective taxonomy, preventing a sole focus on factual reproduction. This approach grants students a degree of control over exam content and teaches them principles of effective question design.

An additional benefit is that students engage in learning for the exam while actively generating questions. Moreover, a strong link exists between the project work and the OBE, as all knowledge pertinent to the project is also relevant for the OBE (Brightwell et al., 2004). An OBE is more authentic to real-world practice, "because professionals of the future will not be able to 'know' all the information needed for competent performance" (Durning et al., 2016, p. 583). The use of AI (e.g., Gemini) in creating realistic case-based tasks further enhances authenticity:

> *"A psychological practice specializing in the treatment of anxiety disorders has noticed that some experienced therapists have a tendency towards burnout symptoms and decreasing self-efficacy in dealing with particularly challenging clients, even though they are highly*

*skilled. They would like support in reflecting on their personal approach and in developing individual coping strategies for their high workload. Would training or coaching be more suitable here? Justify your choice in 2–3 sentences using at least two distinct arguments."*

### 4.3. Teaching and Learning Activities: Focus of the HRD Interventions

Needs assessment is a critical initial phase in HR intervention design, as it is indispensable for ensuring the intervention targets authentic needs or gaps within an organization. More precisely, the information drawn from the needs assessment "can be used to establish priorities for expending HRD efforts, define specific training and HRD objectives, and establish evaluation criteria" (Werner & DeSimone, 2012, p. 27). Within this assessment, methodologies such as strategic/organizational, task, and person analysis can be employed. The latter, person analysis, specifically involves identifying employee competency gaps that may hinder their ability to effectively perform current and future tasks (Bansal & Tripathi, 2017; Werner & DeSimone, 2012).

At this juncture, where supervisory interviews would typically be conducted, the educator intervenes to ensure that the groups conceive and implement a diverse range of HRD interventions. Students are tasked with preparing interview questions relevant to a person analysis. The educator, serving as the interview partner, receives these questions and utilizes AI (e.g., Gemini) to generate responses. This process allows the educator to strategically direct the focus towards specific competencies (e.g., interdisciplinary competencies, perspective-taking and ambiguity tolerance, teamwork and conflict management skills, self-reflection skills related to cognitive patterns). This strategic intervention ensures that student groups develop distinct interventions and, consequently, also participate in different interventions. Students are responsible for independently conducting the other levels of the needs assessment and ultimately establishing the link to the person analysis (e.g., "Are the competencies identified in the task analysis also reflected in the person analysis?"), fostering learner agency and authentic knowledge practices.

### 4.4. Teaching and Learning Activities: PjBL Characteristics

Overall, the course design embodies the core features of PjBL as outlined by Shpeizer (2019), while simultaneously integrating key principles of open pedagogy (Bali et al., 2020; Bovill, 2020; Cronin, 2017; DeRosa & Robison, 2017). An illustration of the PjBL characteristics that are embodied in the course design is displayed in Figure 2. A detailed description follows below.

#### 4.4.1. In-Depth Inquiry

This is reflected in the students needing to conceive, implement, and evaluate a complete HRD intervention. This requires them to delve deeply into the relevant theories, models, and practical considerations of HRD. The sequential nature of the project, with weekly tasks building towards the final intervention and project report, also encourages in-depth engagement.

#### 4.4.2. Authenticity

Authenticity is enacted through the real-world task of designing an intervention for future psychologists, aimed at preparing them for their careers by equipping them with key professional qualifications. Authenticity was defined not solely by the length of the intervention but by the extent to which students engaged in professional tasks that mirror real-world HRD practices. Students complete the full HRD cycle, including conducting a needs analysis, designing an intervention grounded in theory, implementing it for an authentic audience (their peers as future psychologists), and evaluating outcomes.

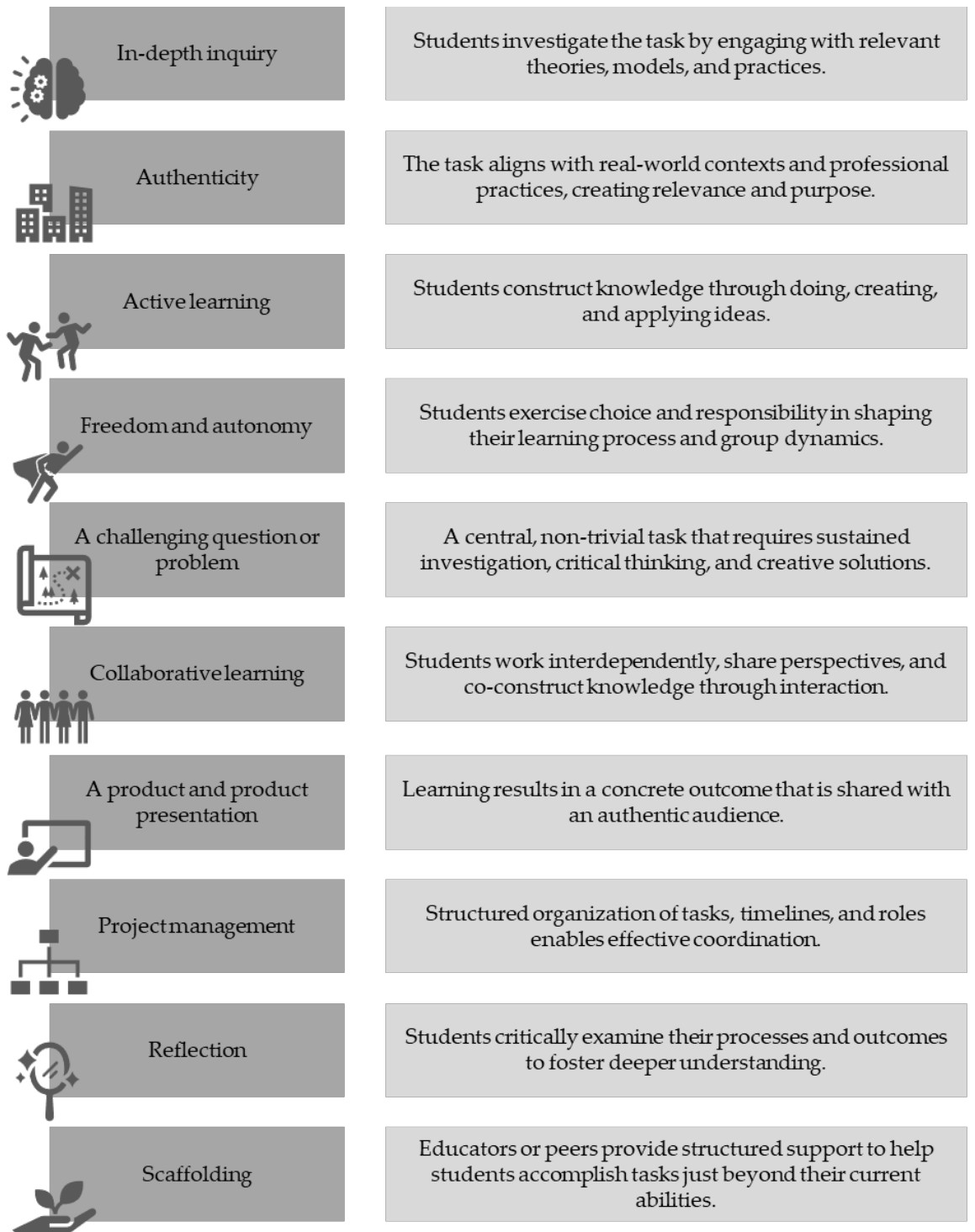

**Figure 2.** Key characteristics of Project-Based Learning (PjBL) embodied in the course design (based on Boss & Larmer, 2018; Shpeizer, 2019; Sulong et al., 2023).

Although the delivery of each intervention is limited to 45 min due to curricular constraints, the semester-long preparation, role rotation (learner, designer, implementer, participant, reviewer), and focus on equipping peers with key competencies ensures that the learning tasks remain authentic in purpose, audience, and process. The educator's role as client transforms traditional hierarchies, creating a learning space based on mutual trust and professional simulation. This aligns with open pedagogy's emphasis on authentic

audience and knowledge application beyond the classroom (Bali et al., 2020; Cronin, 2017; DeRosa & Robison, 2017).

### 4.4.3. Active Learning

Students actively and collectively create knowledge through participation in project conception, implementation, feedback, and even assessment design. The co-creation of exam questions exemplifies openness and transparency in evaluation, and reflects a democratization of assessment processes. The interactive nature of the course sessions also promotes active engagement.

### 4.4.4. Freedom and Autonomy

Learner agency or autonomy is a central feature of the course. Students have control over the design of their HRD intervention and the project report, negotiate group processes through a self-managed team charter, and decide how and when to seek educator input. They are encouraged to coordinate autonomously within their groups. The ungraded nature of the project report also allows for more exploratory and autonomous work without the immediate pressure of grading. The absence of grading for the project encourages risk-taking, creativity, and error-friendly learning—hallmarks of open educational environments.

### 4.4.5. A Challenging Question/Problem

The central challenge of designing a meaningful intervention mirrors authentic complexity. The overarching challenge is to design, implement, and evaluate an HRD intervention that addresses specific professional key competencies of their fellow psychology students. It requires not only disciplinary knowledge, but also inclusivity such as interdisciplinary thinking, stakeholder orientation, and collaborative negotiation—skills emphasized in open and participatory pedagogies.

### 4.4.6. Collaborative Learning

The course structure heavily emphasizes collaborative learning through the formation of small, semester-long groups. Students work together on all phases of the project, from initial conception to the final report. Students rotate roles and give and receive peer feedback. The diversity of contributions and mutual dependency foster inclusive participation and collective ownership.

### 4.4.7. A Product and Product Presentation

The project report serves as a primary tangible product, documenting the entire process of conceiving, implementing, and evaluating the HRD intervention. Furthermore, the implementation of the HRD intervention itself acts as a form of product presentation to the target audience (their fellow students), showcasing their designed intervention in action. Since the interventions are created for real peers with real needs, not just for the educator, the principle of knowledge sharing is reinforced.

Boss and Larmer (2018) also offered a research-and-practice-informed framework for high-quality PjBL and set out six criteria, two of which are not reflected in Shpeizer's (2019) criteria: Project management and reflection.

### 4.4.8. Project Management

The semester-long small group work necessitates project management skills such as collaboration, task division, and coordination. Framing the groups as "service providers" further emphasizes planning and delivery. The weekly structure with content input and project work establishes a natural timeline. Explicitly, the team charter supports defining roles and responsibilities, while the iterative feedback and revision process of the project

report mirrors project management cycles of monitoring and adjustment. The team charter reflects participatory governance. Iterative refinement cycles parallel the cyclical nature of formative development.

### 4.4.9. Reflection

Reflection is embedded through multiple mechanisms: peer review prompts self-reflection—both while evaluating others' work and while reviewing the feedback, educator feedback guides improvement, and the design process of the HRD intervention encourages consideration of its effectiveness. Additionally, creating exam questions necessitates reflection on course content and learning objectives. These activities encourage metacognition and help students critically evaluate their own learning. Reflection here is both an individual and collective process, rooted in transparency and growth.

In addition, Sulong et al. (2023) also outlined several PjBL principles based on, for example, Condliffe (2017), Krajcik and Shin (2014), and Le (2018). One principle, that is not reflected in Shpeizer (2019) or Boss and Larmer (2018), is scaffolding.

### 4.4.10. Scaffolding

Scaffolding in this course is primarily implemented through the educator's role as a facilitator, providing ongoing guidance, structure, and resources to support students' successful completion of their HRD projects. This includes the delivery of relevant theoretical knowledge in each interactive session, which directly informs project work. As client, the educator defines the project's scope and offers feedback on the developing project report.

The collaborative creation of a team charter also provides a foundational structure for group work. While the educator remains available for support upon request, the pedagogical design does not emphasize a gradual reduction in this support, as the complexity of the HRD project requires consistent guidance throughout the semester. The emphasis lies on empowering students to self-regulate and co-navigate complexity within a safely scaffolded environment. This approach aims to empower student autonomy within a consistently supported learning environment.

### 4.5. From Principles to Practice: Synthesizing Open Pedagogy and PjBL

While the previous sections described the principles of open pedagogy and the concrete implementation of PjBL in our HRD course, it is important to explicitly synthesize how these two frameworks interrelate. Open pedagogy provides the *theoretical foundation*, emphasizing values such as transparency, co-creation, learner agency, and authentic knowledge practices. PjBL, in turn, serves as the *methodological vehicle* through which these values can be enacted in practice.

In this course, we understood open pedagogy as setting the direction and guiding principles, while PjBL offered the pedagogical structure to translate these principles into tangible teaching and learning practices. For example, the value of *co-creation* was operationalized not only through student participation in project design but also in their active role in developing OBE tasks. Similarly, the principle of *authentic knowledge practices* was realized through the design of HRD interventions that addressed real-world professional competencies of future psychologists. *Learner agency* was supported by students' autonomy in group work, including their use of team charters, while *democratization and transparency* were embedded in OBE and iterative peer feedback processes.

Equally central was the principle of *collaboration* (i.e., participatory or co-learning). It was realized through stable small groups, role rotation, and peer-to-peer review, ensuring that learning was conceived as a social, participatory process rather than an individual one. The emphasis on *formative learning cycles* was visible in the iterative structure of project development, where different stages involved feedback, revision, and reflection.

*Inclusivity* was fostered both by valuing diverse perspectives within group work and by designing an error-friendly environment that encouraged experimentation without the fear of punitive grading.

By aligning open pedagogy's values with PjBL's structural characteristics, we aimed to create a coherent learning environment in which students were not only recipients of content but also active participants in shaping knowledge, practice, and assessment. This integration was also intended to shift the educator's role from transmitter of knowledge to facilitator, co-learner, and mentor.

This conceptual synthesis forms the basis for the following evaluation. Rather than viewing outcomes only through the lens of improved competencies, we also assess how students and educators experienced openness, participation, and authenticity. In doing so, the evaluation reflects not only the effectiveness of the course design but also its potential contribution to cultural change in higher education toward more participatory and transparent practices.

## 5. Evaluation of Novel Course Implementation

In line with the third phase of the DBR process—implementation and evaluation—we conducted an empirical assessment of the newly designed course format. This evaluation aimed to determine the extent to which the intervention achieved its intended learning outcomes and supported a shift toward participatory, practice-oriented, and open learning environments. It also served as a basis for reflection and iterative refinement in future course cycles (McKenney & Reeves, 2018). The course evaluation combined quantitative and qualitative approaches to capture both students' learning progress and their perceptions of the redesigned course. This study should be understood as a preliminary exploratory case study within the first DBR cycle, designed to yield insights for ongoing course development rather than to produce broadly generalizable results.

### 5.1. Participants and Procedure

This pedagogical transformation was implemented across two courses within the Bachelor of Psychology program at Kiel University, Germany in the summer of 2025. One course comprised 15 students, while the other had 16 students. At the beginning of the semester, students were informed about the new course format and invited to participate in a pre-test survey (t1). We obtained informed consent for participation and publication, and the study was carried out following the rules of the Declaration of Helsinki and is in accordance with the General Data Protection Regulation (European Union Regulation, 2016). Of the 31 enrolled students, 28 participated in the initial pre-test. The average age of participants was 25.14 years ($SD = 7.27$). The courses were predominantly attended by female students (82.1%). The average academic semester of participants was 6.21 ($SD = 0.83$). Therefore, most of the participants were third-year students. Most of the students also stated that they wanted to apply for a master's degree in psychology with a clinical focus after their bachelor's degree (82.1%).

Following the course, students were again invited to participate in a post-test (t2), in which $n = 23$ students took part. Participation in the surveys was voluntary and without advantage, which is why the surveys had to be kept short. Unfortunately, due to the voluntary nature of the surveys, we were unable to ensure that all students participated. We also had trouble matching the data from the first survey with the second for some students, as a few hadn't recorded their personal ID from the first survey, which was necessary for assignment. Thus, we could only match data for $n = 16$ persons.

### 5.2. Measures, Data Analysis, and Results

5.2.1. Pre-Test

To establish a baseline for evaluating the impact of the course design, we administered a pre-course survey at the beginning of the semester (t1).

5.2.1.1. Quantitative Data

To assess students' knowledge and competencies, we have translated our learning objectives and desired practical qualifications into several items, which should not be viewed as multi-item scales but be analyzed on an item level. As these items were designed for descriptive monitoring rather than psychometric validation, reliability coefficients are not reasonable. Accordingly, results should be interpreted as exploratory and descriptive indicators of change rather than as standardized psychometric measures.

Students were asked to self-assess their competence regarding the intended learning objectives (see Table 3; called learning objectives). Furthermore, students were requested to evaluate their application competence, specifically the extent to which they felt capable of implementing HRD interventions in practice (see also Table 3; called professional qualification).

**Table 3.** Descriptive statistics and results of the Wilcoxon signed rank test for the items on learning objectives and professional qualification assessed at t1 and t2.

| Variable | Item | t1 (Pre-Test) | | | t2 (Post-Test) | | | Wilcoxon Signed Rank Test for Matched Data ($n = 16$) | Effect Size r (Z/$\sqrt{N}$) |
|---|---|---|---|---|---|---|---|---|---|
| | | $n$ | $M$ (SD) | $Md$ | $n$ | $M$ (SD) | $Md$ | $p$-Value (One-Tailed) | |
| Learning objectives | I have knowledge of how HRD interventions should be implemented from a scientific perspective. | 28 | 2.57 (0.96) | 2.5 | 23 | 4.52 (0.59) | 5.0 | <0.001 | −0.90 |
| | I have knowledge of the various methods that can be applied in HRD interventions. | 28 | 2.57 (0.84) | 2.5 | 23 | 4.30 (0.63) | 4.0 | <0.001 | −0.89 |
| | I have knowledge of learning theories to understand how people learn. | 28 | 4.04 (0.51) | 4.0 | 23 | 4.74 (0.54) | 5.0 | 0.002 | −0.90 |
| | I have knowledge of the catalysts and barriers to the effectiveness of HRD interventions. | 28 | 1.79 (0.79) | 2.0 | 23 | 4.09 (0.85) | 4.0 | <0.001 | −0.89 |
| | I possess the ability to practically apply my knowledge of HRD interventions and develop practical solutions for real-world problems. | 28 | 2.29 (0.94) | 2.0 | 23 | 3.96 (0.64) | 4.0 | <0.001 | −0.85 |
| Professional qualification | I feel capable of developing and implementing HRD interventions in practice. | 28 | 1.79 (0.74) | 2.0 | 23 | 3.35 (0.98) | 3.0 | <0.001 | −0.89 |
| | I could explain to someone else how HRD interventions should be developed and implemented. | 28 | 2.07 (0.94) | 2.0 | 23 | 3.91 (0.90) | 4.0 | <0.001 | −0.89 |
| | I am qualified for the occupation as an HR developer. | 28 | 1.68 (0.67) | 2.0 | 23 | 2.74 (0.86) | 3.0 | 0.001 | −0.89 |
| | I feel well-prepared for a role as a psychologist in HRD. | 28 | 1.68 (0.61) | 2.0 | 23 | 3.35 (1.11) | 3.0 | <0.001 | −0.89 |

*Note.* All items were administered on a 5-point Likert scale ranging from 1 (*do not agree at all*) to 5 (*fully agree*). HRD = Human Resource Development.

### 5.2.1.2. Qualitative Data

Open-text responses were collected to capture students' expectations and needs. These responses were reviewed systematically and inductively categorized into recurring themes (e.g., expectations of practical application, career orientation, interactive learning). Within each theme, representative anchor quotes were selected to illustrate the breadth of student perspectives, and the frequency of mentions was noted to indicate prevalence. The full set of anchor examples is provided in Appendix A, Table A1 to enhance transparency. These responses served both as needs analysis and as a reference point for assessing alignment between student expectations and the intervention design.

The results showed that, in addition to acquiring theoretical knowledge and linking it to previous knowledge ($n = 17$; "I want to leave the course with a good overview of the HRD process"), most students emphasized the importance of application-oriented learning experience that bridges theory and practice ($n = 20$; I would wish that we work in a very practice-oriented way, 'learning by doing', so to speak"). Students expressed a desire for hands-on tasks, case studies, and real-world examples, and a "learning by doing" approach.

An emphasis was also placed on career orientation ($n = 7$; "I wish for insights into what it's like to work in work and organizational psychology"), with students eager for industry connections and experience reports from professionals (e.g., guest speakers). They also valued an interactive and engaging environment and balancing content with group work and dialog ($n = 16$; "The course should be lively, many interactions"). Additionally, they appreciated transparency and a clear course structure with defined goals and evaluation. Workload management was also a concern, particularly given bachelor's thesis commitments, leading to a preference for manageable time investments and practical assessments ($n = 9$; "clearly defined expectations for the performance to be delivered by the students").

In essence, students were looking for an interactive, application-oriented learning experience that bridges the gap between academic theory and professional practice, all within a supportive and well-structured environment. These findings informed the final course structure and are consistent with open pedagogical principles such as learner agency, authentic knowledge practices, and transparency.

Finally, students had the opportunity to indicate what additional elements they would require feeling better prepared for a career in HRD (for anchor examples, see Appendix A, Table A1, Needs). Students' responses suggested a need for theoretical knowledge ($n = 17$; "Currently, I still lack a lot of knowledge, which I am sure I will gain in the course"), methodological and practical competence ($n = 12$; "Knowledge about the techniques and methods applied in HRD"), and room for practicing ($n = 13$; "To become active myself and practice in a practical way. So far, it has mostly just been theories...."). They emphasized bridging the theory-practice gap by applying theoretical knowledge to real-world scenarios through concrete case examples and step-by-step guidelines. They also required deepened and current theoretical foundations and desired real-world expertise from professionals to understand the practical nuances of HRD, along with a comprehensive understanding of implementation processes.

### 5.2.2. Post-Test

After the semester, students completed a second survey (t2) with parallel items to assess development in their knowledge, competencies, and perceptions of the course format. This post-course evaluation also functioned as an embedded reflection tool in the DBR cycle, enabling evidence-informed refinement for future iterations of the intervention.

5.2.2.1. Quantitative Data

Wilcoxon signed rank tests, chosen due to non-normal data distribution, revealed significant improvements for all learning objectives and application competencies for students who provided data at both time points (Table 3). Effect sizes (r for Wilcoxon signed rank tests) were large, indicating that the observed differences are not only statistically significant but also practically meaningful.

Regarding the evaluation of the course and course's concept, students were asked to rate several questions on, for example, the organization and structure, interest and motivation, cognitive activation, support and feedback, and supervision (Table 4). The items used correspond to our university's internal standard evaluation tool. The items do not represent multi-item scales but are interpreted at the item level.

**Table 4.** Means and standard deviations of the evaluative items assessed at t2.

| Category | Item | M (SD) |
|---|---|---|
| Organization and structure | The course was clearly structured in terms of content. | 4.09 (1.06) |
| | The instructor used the available time effectively. | 4.55 (0.67) |
| Clarity and understandability | The instructor clarified the learning objectives of the course. | 4.23 (0.92) |
| | The instructor expressed herself clearly and understandably. | 4.14 (0.99) |
| Workload and demands | The pace of the course was appropriate for me. | 4.41 (0.73) |
| | The scope of the course material was appropriate for me. | 3.82 (1.26) |
| Interest and motivation | The instructor made the course interesting. | 4.14 (0.71) |
| | The course promoted my interest in learning content. | 3.68 (1.09) |
| Cognitive activation | The instructor encouraged me to think, e.g., through open questions. | 4.32 (0.65) |
| | The course promoted engagement with the learning content. | 4.73 (0.46) |
| Conceptual understanding | The instructor provided vivid examples that contributed to understanding the learning content. | 4.82 (0.39) |
| | The instructor repeatedly made connections to previously taught learning content. | 4.27 (0.70) |
| Support and feedback | The instructor asked meaningful questions to check understanding. | 4.18 (1.01) |
| | The instructor provided feedback on the progress of the learning process. | 4.18 (0.96) |
| Classroom climate | In case of disruption, the instructor reacted confidently. | 4.00 (1.15) |
| | The instructor contributed to a respectful interaction in the course. | 4.64 (0.49) |
| Supervision | I felt well supervised by the instructor during class time. | 4.50 (0.80) |
| | Even outside of class time, I felt well supported by the instructor with my concerns. | 4.59 (0.59) |
| Course design | The instructor communicated expected examination criteria transparently. | 3.91 (1.27) |
| | The instructor excessively delegated the conveying of knowledge to students (e.g., too many student presentations). | 2.05 (1.17) |
| Comparison to more traditional concepts | I feel that, through the course concept, I gained a deeper understanding of the topic than would have been the case with more traditional concepts. | 4.23 (0.87) |
| | This course concept motivated me more to participate than previous course concepts. | 3.55 (1.06) |
| | In this course, the practical relevance was higher than in other courses. | 4.64 (0.58) |
| | Compared to others, this course concept fostered my feeling of autonomy and personal responsibility for my learning. | 4.14 (0.83) |
| Development of competencies and skills | Through the project work, I was able to develop or acquire various competencies and skills: | |
| | Critical thinking | 3.55 (1.10) |
| | Problem solving | 4.00 (0.76) |
| | Teamwork, collaboration | 4.00 (0.93) |
| | Creativity | 3.45 (1.10) |
| | Self-efficacy expectation | 3.59 (1.05) |
| | Communication | 3.73 (0.94) |
| | Self-management | 4.05 (0.65) |
| | Project management | 4.18 (0.59) |

*Note.* All items were administered on a five-point Likert scale ranging from 1 (*do not agree at all*) to 5 (*fully agree*).

Many of the means were in the upper range of the Likert-scale, which can be considered good. For example, the highest mean was 4.82 (*SD* = 0.39) for the item "The instructor

provided vivid examples that contribute to understanding the learning content" in the dimension "conceptual understanding." Slightly lower ratings (<4.0) were observed for the items "The scope of the course material was appropriate for me" ($M = 3.82$, $SD = 1.26$), "The course promoted my interest in learning content" ($M = 3.68$, $SD = 1.09$), "The instructor communicated expected examination criteria transparently" ($M = 3.91$, $SD = 1.17$), and "This course concept motivated me more to participate than previous course concepts" ($M = 3.55$, $SD = 1.06$). However, the higher standard deviation (>1.0) for these items also shows that the responses differed more than for most other items.

In addition, students stated that they had developed several skills and competencies at least to a moderate extent (e.g., problem solving, teamwork and collaboration, self-management, project management; see Table 4).

Lastly, the examination grades were utilized as an additional indicator for the effectiveness of the course concept. The grades in Germany vary between 1.0 (very good) and 4.0 (sufficient). A grade lower than 4.0 simply means failing. The grades are further divided into, e.g., 1.3 and 1.7. The average grade in this course was 1.3 ($SD = 0.6$), with a median of 1.0. Twenty-three people passed the exam with a 1.0, three with a 1.7, three with a 2.0, one person with a 2.3 and one person with a 3.7. All students passed. This result can thus be classified as very good.

In addition, students should also give the course a final overall grade. The course received an average grade of 1.65 ($SD = 0.39$, range = 1.0–2.3) from the students, which can be classified as good.

### 5.2.2.2. Qualitative Data

Additionally, they were prompted to reflect on the course concept, identifying both positive aspects and areas for improvement, and what additional elements necessary for them to feel adequately prepared for a career in HRD (for examples, see Appendix A, Table A2).

Regarding the positive aspect, students mainly emphasized the practical orientation and examples ($n = 13$), that the project was not graded ($n = 4$), that there were no graded oral presentations ($n = 4$), and they also emphasized the OBE ($n = 3$). Three students highlighted the structure of the course and the concept. Two students emphasized the importance of interaction and teamwork. Two students said that they experienced the course interesting and varied.

Regarding the limitations, most students criticized the heavy workload ($n = 10$) and the unclear communication regarding expectations and requirements ($n = 8$). Two students criticized the structure of the course, the short time between content delivery and application, the high proportion of lectures given by the educator, and the feedback culture.

Regarding what students need to effectively work in HRD, fifteen stated that they need more hands-on experience. Three students also indicated that further courses on this topic would be helpful. Two students expressed a wish to receive feedback on the implementation of the intervention and to gain insight into practice through others.

These findings provided empirical grounding for the subsequent DBR reflection phase and informed targeted design improvements for future iterations.

### 5.3. Practitioners' Reflection on the Opportunities and Challenges of the Novel Course Design

As the final phase of the DBR process, we critically reflected on the implementation from the educator's perspective. This phase—Reflection to Inform Redesign (McKenney & Reeves, 2018)—enabled us to identify both the perceived opportunities of the new learning format and challenges that emerged in practice. These insights directly informed adjust-

ments for future iterations and contribute to broader discussions about open, participatory teaching in work and organizational psychology.

5.3.1. Opportunities

From our perspective as reflective practitioners, the implementation of the new learning format revealed a range of benefits that extended beyond conventional teaching gain. PjBL offered numerous advantages that traditional formats often lack. Perhaps the most significant benefit we experienced was the changed role of the educator. In other concepts, the primary focus was on knowledge transfer. With PjBL, we also faced the demand of challenging, supporting, guiding, and mentoring students. The opportunity for students to directly apply what they learned in group work during each session also gave us insight into group processes, allowing us to directly observe how students reflected on and applied their new knowledge. This enabled us to better fulfill our role as facilitators, focusing on guiding, coaching, and providing individualized feedback.

This shift also highlights the persistent tension between traditions in higher education and facilitative, learner-centered pedagogies. It illustrates how institutional expectations of efficiency and content delivery can conflict with the relational and process-oriented roles that open pedagogy requires.

Another positive aspect we experienced was that not only did students become learners, but we as educators did too. This led to co-learning or co-creation in learning and teaching, where we and learners actively participated in constructing knowledge and shaping the learning experience (Bovill, 2020; Kaminskiene et al., 2020). It includes shared ownership, mutual learning where we as educators also learn from students, active participation from all parties—encouraging dialog, experimentation, and critical reflection—flexibility and adaptability, and the focus on the process where the journey of discovery and the collaborative construction of understanding is valued (Bovill, 2020; Kaminskiene et al., 2020). Moving towards application and away from pure theory required us to engage even more intensely with the subject matter and potential pitfalls. Through continuous feedback from students regarding possible issues, we could respond flexibly, refining tasks or providing helpful examples as needed. This not only fostered learning and reflection on both the pedagogy and content effectiveness but also creativity and innovation in teaching.

Co-learning also challenges established hierarchies in academia, redistributing authority in ways that may unsettle both students and educators. As noted in open pedagogy literature (Bali et al., 2020; DeRosa & Robison, 2017), such democratization of knowledge production can advance equity but also requires institutional cultures to value co-creation as a legitimate form of teaching and scholarship.

Finally, we experienced the use of AI to be highly beneficial. AI served as a service tool for creating case studies used in sessions and helped generate realistic, practice-oriented OBE questions that required higher levels of learning. Most helpfully, AI assisted in answering student interview questions during the needs analysis—allowing specific needs, previously defined and narrowed down by clear prompts, to be revealed to students. However, it also raises questions about the balance between technological augmentation and human judgment in education. This tension aligns with broader debates on digital transformation in higher education (Alfredo et al., 2024), where efficiency gains risk overshadowing pedagogical intentionality if not critically managed.

Overall, this was a very positive and enriching experience. However, quantitative, qualitative, and reflective evaluations on the project have also identified areas where further refinement can be made.

5.3.2. Challenges and Lessons Learned

We experienced that implementing a novel pedagogical approach like PjBL inevitably presents both anticipated and unforeseen challenges, offering valuable insights for future iterations. Based on our experience, several key areas warrant reflection and adjustment.

Firstly, the mandated participation in the course, combined with the curriculum's requirement for a graded outcome, posed a challenge on us to fostering intrinsic motivation in students. While participation is essential, relying solely on compulsory engagement can inadvertently diminish students' innate drive to learn (one student, for example, stated that she/he needs "interest in HRD" to feel qualified to work in the context of HRD). This reflects a fundamental tension between student autonomy and curricular structures that mandate participation. It resonates with the self-determination theory (Ryan & Deci, 2000), where externally imposed requirements can undermine intrinsic motivation, raising broader questions about how innovative pedagogies can thrive within rigid institutional frameworks. We feel that this leads to a fundamental tension that requires careful consideration in future course designs.

One way to deal with this situation, for example, would be to focus on case studies and possibly on the interventions in the field of clinical psychology, as most students were professionally interested in this area. In this way, the importance of HRD, especially for this specific field, can be emphasized, and students can better understand why they need to take this course and what the content is relevant for. Also, drawing parallels, for example, to the process of psychotherapy (e.g., clarification of needs, goal definition, development and application of tailored methodology, and evaluation) can help students recognize the "bigger picture" and learn to appreciate the value of HRD. Since some will certainly also aim for leadership positions, whether in the field of clinical psychology or not, the necessity of knowledge about HRD should also be emphasized.

Secondly, we experienced that evaluating the efficacy of the PjBL concept itself proved difficult (see also Shpeizer, 2019). Students—psychology students in particular—already managing demanding schedules, exhibited low motivation for participating in additional surveys and data collection beyond the core course requirements. One possibility would be to explicitly allocate time in the first and last sessions for participation in the evaluation, even within a tight schedule. Furthermore, establishing a comparable control group for robust evaluation was challenging, as our pedagogical philosophy prioritizes offering all students the same high-quality learning opportunities. This challenge highlights a common dilemma in educational innovation: balancing rigorous evaluation with equitable learning experiences. It exemplifies the "double role" of educational design research as both a process of teaching improvement and a place of scientific research, often requiring compromises between rigor and equity.

We also experienced that the introduction of a new concept also demands understanding and flexibility from students regarding educator uncertainties. As educators, we are also learning through experimentation and reflection. Students must therefore understand that initial milestones, such as submission deadlines, may not be rigidly fixed and that the schedule might evolve to accommodate emerging needs or external circumstances. This necessitates that we cultivate the competence to manage uncertainties and student inquiries, fostering an atmosphere of calm and reassurance.

We also observed that the inherent freedom and experimental nature of PjBL, while beneficial, can also induce resistance and uncertainty among students, potentially leading to anxiety or feelings of overwhelm. Clear communication of requirements, milestones, and deadlines is paramount to mitigate these uncertainties and provide a stable structure for both students and instructors. However, this clarity must be balanced with flexibility, as unforeseen external circumstances may always necessitate adjustments to schedules or

content. For example, to allow students to meet master's program application deadlines, we had to reschedule the course exam earlier.

Furthermore, we became aware during the semester that students were simply memorizing the slides for the exam and repeatedly asked if every minor detail was relevant for the exam. Consequently, we changed the exam, originally designed as a closed-book assessment, but converted on short notice (two weeks prior) to an open-book format to test understanding and application rather than rote memorization of content. Many students evaluated the OBE positively, although some expressed dissatisfaction with the short-notice change. These changes thus required considerable flexibility from both educators and students.

For future implementations of this concept, we recommend clearly defining requirements and expectations for all assessment components (graded or not). We also advocate for flexibility in responding to changes even during the course, to prevent phenomena like overcommitment. Conclusively, the need for flexibility highlights a broader systemic tension: innovation demands adaptive responsiveness, yet higher education often values stability and standardization. This illustrates how institutional cultures of predictability can collide with pedagogical designs that embrace emergence and iteration.

For us, another critical learning point emerged regarding the time allocated for student-designed HRD interventions. Each group had only 45 min to implement their developed intervention with their peers. This timeframe unfortunately does not reflect the demands of real-world practice, where interventions can span days, weeks, months or even years (Swanson, 2022). It is important to acknowledge this disparity and clearly communicate expectations regarding the scope and scale of the interventions within the given time constraints.

It is also notable that many students criticized the workload. Even though the workload aligned with the curricular requirements, many perceived it as too high. Criticism was directed not only at the workload itself but also at the one mandated by the curriculum. This also aligns with our first point of criticism regarding the lack of intrinsic motivation, as many students showed little interest in the topic and were thus less motivated to put much effort into the course. At present, we see no further approaches to address this dissatisfaction.

Finally, the observed lack of intrinsic motivation among students for course participation was further evidenced by their reluctance to attend the final session. This session, designed to feature guest speakers sharing real-world insights into HRD practices (according to the desire expressed by several students prior to the course on "Career orientation and industry connection"), ultimately had to be canceled. The anticipated attendance did not justify the time and resource commitment of the guest speakers. Although attendance was compulsory, students were permitted to miss up to two sessions without consequences.

This situation highlights a gap between the intended learning experience and actual student engagement, particularly with regard to the perceived value of practical insights from industry professionals. It also highlights a broader issue in educational innovation: creating authentic engagement requires not only offering opportunities but also cultivating the dispositions to value them. For us, this discrepancy is surprising, given that many students have expressed a desire for practical insights. It was somewhat frustrating for us as we attempted to provide students with meaningful insights into the field. We therefore decided to invite the guest speakers for an evening session and open this session to all psychology students, to truly reach those who are interested in the topic.

An important insight from these reflections is that we are constantly navigating the tension between educational innovation and institutional requirements and constraints. While we are highly motivated to design and implement new, stimulating, and challenging concepts, we must constantly operate within the framework of existing guidelines, which

impose significant constraints. We have observed instances where, despite adherence to these guidelines, students' perceptions and evaluations differ from the intended outcomes, as was the case with the perceived workload. Although the workload was consistent with curricular requirements and was an intentional design element, many students subjectively assessed it as being excessive. Therefore, our task as educators is to identify a suitable balance that can satisfy the interests of all three parties involved.

In conclusion, the most important lesson we are taking away from this experience is that you cannot please everyone, and you should free yourself from that expectation. The qualitative data, in particular, clearly showed how differently students perceived this new concept. While some noted they had no suggestions for improvement, experiencing the course successful and helpful, others criticized things like the workload or communication. Different preferences also emerged regarding the project work: many evaluated it valuable that it was not graded, while others would have preferred it to be graded. There will always be a few whose needs you cannot meet—and you do not have to.

## 6. Discussion

The present study critically evaluated the implementation of PjBL as a transformative approach in an undergraduate HRD course in psychology, highlighting not only its pedagogical effectiveness but also its contribution to open practices and the cultural transformation of academic education. In line with the guiding research questions, this discussion interprets (1) how PjBL, informed by open pedagogy, can be designed and implemented in HRD courses to foster openness and student participation, and (2) what advantages, challenges, and cultural implications arise for students and educators. Ultimately, this DBR sought to present promising practices and preliminary data addressing how PjBL can be effectively designed and implemented in HRD and similar courses and explore its perceived advantages in this context.

In the initial phase of our DBR cycle, reflecting on the status quo and assessing student expectations, we found that students expressed a clear preference for an authentic, practice-oriented HRD course that connects academic theory with real-world applications. Such expectations resonate with the broader call for transparency, inclusivity, and participatory structures in higher education, which are central to the open practices agenda (Bali et al., 2020; Cronin, 2017; Lambert & Czerniewicz, 2020). The principles of transparency, structure and communication were also deemed to be of significance. In addressing these concerns, the previous format was transformed into PjBL.

Through the principles of constructive alignment (Biggs & Tang, 2011) and open pedagogy (Bali et al., 2020; DeRosa & Robison, 2017), as well as through various characteristics of PjBL (Boss & Larmer, 2018; Shpeizer, 2019; Sulong et al., 2023)—such as authenticity, active and collaborative learning, freedom and autonomy, and reflection—we were able to promote self-directed and student-centered learning. The present study adopted the DBR approach (McKenney & Reeves, 2018), which involved the following sequence of steps: analysis of the status quo, development of a new concept, implementation and evaluation of this concept, and reflection on the opportunities and challenges from a practitioner's perspective. This iterative process illustrates how openness can be embedded as both a pedagogical practice and a research principle, thereby linking teaching innovation with cultural change.

Following a comprehensive evaluation, it was determined that the learning objectives were predominantly fulfilled, and there was a substantial enhancement in the students' professional qualifications. Furthermore, students reported enhancements in their interdisciplinary competencies and skills, including project and self-management, teamwork and collaboration, and problem solving. This finding is corroborated by research on PjBL

(Guo et al., 2020; Smolyaninova et al., 2021; Sulong et al., 2023; Zhang & Ma, 2023). With respect to the expectations articulated by students regarding a satisfactory course on HRD, it is considered that many of these have been fulfilled. Particularly noteworthy is that students valued the participatory structures and practical orientation, which align closely with open education principles of co-creation, collaboration, and authentic knowledge practices (Bali et al., 2020; Bovill, 2020; Cronin, 2017; DeRosa & Robison, 2017). The successful transmission of theoretical knowledge is reflected in the achieved learning objectives and OBE grades.

From our perspective, an interactive and engaging learning environment was also present; students, for example, stated that they positively experienced the exercises, as well as interaction and teamwork. The high ratings for the evaluation dimensions of interest and motivation, in addition to the comparison to more traditional concepts, also support this resume. The thorough evaluation of the course and its underlying concept, in conjunction with the favorable grades awarded by students, signifies a successful and well-received transformation of our pedagogical approach. At the same time, these findings demonstrate how openness in design, assessment, and collaboration can enhance both student engagement and perceived relevance, thereby contributing to cultural transformation in higher education A synthesis of practitioner reflections and student feedback yielded the following key findings.

From an educational perspective, a shift in role has been identified among educators. The transformation of the educator role that has been observed—moving towards that of facilitator, co-learner, and mentor—is in alignment with open pedagogy principles. This shift exemplifies how open practices can challenge existing academic hierarchies and foster more equitable participation in teaching and learning (Bali et al., 2020; DeRosa & Robison, 2017).

Furthermore, technological innovation can serve as a pivotal force in the evolution of modern pedagogy. AI can be used not only to create authentic case studies but also in close collaboration to develop appropriate OBE questions that are precisely tailored to the course content and promote application, evaluation, and transfer rather than mere rote memorization. This approach is extremely resource-efficient (Budhwar et al., 2022; Golgeci et al., 2025), as the time required to create such questions (and possible answers) is significantly reduced. Furthermore, iterative improvement cycles could be conducted using AI, resulting in questions of very high quality and practical relevance.

Beyond the use of AI for generating realistic case studies and co-designing exam questions, future applications could also include personalized feedback and learning analytics. For instance, AI could provide students with individualized feedback on draft project reports from different perspectives and stakeholders or simulate varied organizational contexts for needs analyses, thereby extending authenticity and deepening practice-oriented engagement. Moreover, AI-driven analytics could help educators identify patterns in student learning processes, offering opportunities for more targeted scaffolding. While these features were not implemented in the present study, they represent promising avenues for enhancing both the authenticity and personalization of PjBL in higher education. We therefore advocate that educators embrace the challenges of digital transformation and engage with AI. How can AI enrich my teaching? Where can I use it, and how does it support me (Golgeci et al., 2025)? Framed through the lens of open practices, AI represents not only a technical aid but also a resource for creating more transparent, accessible, and participatory learning environments.

However, our own reflections and feedback from students also revealed several challenges. In summary, we identified challenges in that compulsory participation and mandatory grading had a negative impact on students' intrinsic motivation. This perception was also evident in student assessments of the workload: although it was in line with the

curriculum, some students complained about it and noted that other things were currently more important to them. Such tensions reflect structural and institutional barriers that can limit the adoption of open practices (Brandenburger, 2022), particularly when curricular requirements conflict with student autonomy and intrinsic engagement.

We therefore concluded that (1) the relevance of the subject should be clarified to students, for example, by focusing case studies on clinical–psychological areas, which are most likely to be of interest to most students. Through this interdisciplinary connection, more students could be reached and their awareness of the importance of the subject could be raised. (2) In addition, it is advisable to reflect with the students on what would help them increase their motivation and engagement.

At the same time, it is important to acknowledge that the inherent freedom and experimental nature of PjBL, while beneficial for fostering autonomy and creativity, can also challenge students by increasing cognitive load and reducing perceived structure. As Kirschner et al. (2006) emphasized, minimal guidance during instruction may overwhelm learners' working memory, particularly when they are still novices within a given domain.

Our findings resonate with this perspective: the openness of PjBL may have induced resistance and uncertainty among students, occasionally leading to anxiety or feelings of overload. To mitigate these effects, scaffolding—including gradually fading assistance—, transparent communication, and continuous feedback are essential. Providing clear milestones, explicit expectations, and timely formative guidance helps manage cognitive demands while preserving the participatory and self-directed qualities of open pedagogy. In this sense, our experience confirms that openness and structure are not mutually exclusive but must be intentionally balanced to reduce cognitive overload and sustain intrinsic motivation for independent learning.

Building on this insight, we emphasize the importance of clear communication and transparent requirements. By applying this course concept for the first time, we noticed a few aspects in communication that could be improved in the future: (1) clear deadlines, (2) OBE format from the outset, (3) not inviting guest speakers during course time shortly before the exams, and (4) coordinating the exam date with external deadlines. These improvements go beyond technical adjustments; they represent necessary steps in embedding openness, transparency, and trust into course culture. More broadly, the findings show how PjBL, guided by open pedagogy, can contribute to sustainable cultural change in higher education while also revealing the institutional barriers that must be addressed to realize the full potential of open practices.

## 7. Limitations

Even though the present study provides many helpful practical experiences that can lead to improvements in teaching, the study itself, regardless of the subject matter examined, is not without limitations.

We first note the limited validity of the evaluation. For a robust and reliable evaluation, several conditions are necessary (Tamkin et al., 2002). (1) Beyond pre- and post-measurements, a follow-up assessment is needed to gauge temporal transfer of knowledge. (2) To control both intervention-independent (external) and intervention-dependent effects, it is essential to incorporate control and comparison groups. This is the only way to confirm that the intended changes, such as in knowledge, are attributable to the course concept itself. We did not compare our findings with other HRD courses, nor did we supervise students who might participate in the course the following year. (3) While we measured learning and feedback on various levels, it would also be beneficial to assess and evaluate actual professional application skills, perhaps even within a real-world work setting (i.e., behavioral evaluation).

Additionally, we emphasize that the measures used were single items that were self-developed or adapted from institutional evaluation, which provide useful descriptive insights but do not allow for reliability and validity analysis. Accordingly, findings should be interpreted as exploratory. Nevertheless, we would like to explicitly emphasize that we captured and evaluated not only the objective acquisition and application of knowledge, but also the subjective development of competence and the evaluation of the concept itself. This allowed us to assess not only the output but the process itself, which should be highlighted as a positive.

A further limitation concerns the scope of the implemented HRD interventions. Each student-designed intervention could only be conducted within a 45 min session, which does not reflect the duration or complexity of HRD practice in professional contexts, where interventions can span days, weeks, months or even years (Swanson, 2022). We therefore caution against equating this format with full-scale workplace training. At the same time, authenticity in this project was operationalized less through intervention length and more through the process of engaging with the complete HRD cycle—from needs analysis to design, implementation, and evaluation—within a realistic professional context and with an authentic audience (future psychologists). We acknowledge this constraint while maintaining that the focus on process, purpose, and role-taking ensured that the learning tasks retained their authenticity.

Another limitation of the study is the selection and size of the sample. Two courses were evaluated, which were attended primarily by students who were forced to choose the course. It would have been helpful to compare all HRD courses, including those with voluntarily participating students. This would not only have increased the sample size but also allowed for the integration of comparison groups, potentially even with different concepts, into the evaluation design. Furthermore, the sample was limited to predominantly female young-aged students who wanted to apply for a master's degree in psychology with a clinical focus after their bachelor's degree, which also constraints generalizability. However, since we had no influence on the composition of the sample in this setting, this is an issue that can only be addressed to a limited extent. A plausible strategy involves the inclusion of confounding variables in the analytical framework. A key research question would be to what extent students' pre-existing interests, prior knowledge, and intrinsic motivation affect their engagement and evaluation outcomes. This line of inquiry would also address the need for differential support or tailored supervision for specific cohorts of students.

## 8. Practical Implications for Educational Practitioners

Many of the considerations relevant to practice have already been discussed in the preceding reflection (Section 5.3) and discussion (Section 6). What follows is therefore a synthesis of the main points, formulated as practical implications for educators who wish to adopt PjBL in line with open pedagogical principles.

One central implication is the importance of transparent communication. Clear instructions, openly shared assessment criteria, and easy access to resources reduce uncertainty and enable students to focus on substantive aspects of their projects. At the same time, educators must strike a careful balance between autonomy and guidance. While student agency, collaboration, and co-creation are key, formative learning cycles, intermediate milestones, and role rotation provide necessary scaffolding and help to ensure equitable participation.

The findings also underline the motivational potential of authentic learning tasks. By linking projects to professional practices, as in the design of HRD interventions, students perceive their work as relevant and take greater ownership of their learning. Equally, collaboration should be understood not only as an organizational necessity but as a resource

for learning. Stable groups, team charters, and peer review processes foster inclusivity and integrate diverse perspectives into the learning process.

Finally, the use of generative AI opens new opportunities for authenticity in case design and assessment. Its integration, however, requires careful framing to align with learning objectives and to ensure academic integrity.

Taken together, these implications provide a concise summary of how the insights from our study can be translated into actionable strategies. They demonstrate how open pedagogy, enacted through PjBL, can be harnessed to create participatory, transparent, and authentic educational environments that prepare students for the complexities of professional practice.

## 9. Conclusions

This study demonstrated that PjBL, embedded in a DBR framework and guided by open pedagogy, can effectively transform teaching and learning in undergraduate psychology education. By shifting from lecture-based instruction to authentic, co-created tasks, students significantly advanced their theoretical knowledge, practical competencies, and professional confidence. Educators experienced a parallel transformation, moving into roles as facilitators and co-learners, while also exploring the potential of AI to enrich authenticity and efficiency.

Beyond these immediate learning outcomes, our findings underline the cultural impact of open practices in higher education. Openness in curriculum design, assessment, and collaboration fostered transparency, inclusivity, and shared ownership of knowledge (Bali et al., 2020; Cronin, 2017; DeRosa & Robison, 2017). Students valued the participatory structures, though the process also revealed challenges in workload management, communication, and sustaining motivation—reminders of the institutional and cultural barriers that still shape academic learning environments. Addressing these barriers requires both structural support and a shift in mindset toward valuing co-creation and equity in education.

We conclude that integrating PjBL with open pedagogical principles not only empowers future HR professionals but also contributes to the cultural transformation of academia toward openness, sustainability, and innovation. By cultivating participatory, student-centered learning environments, such approaches can bridge the gap between research and education, strengthen interdisciplinary collaboration, and support a more equitable and resilient academic culture.

**Author Contributions:** Conceptualization, S.K. and M.B.; methodology, S.K. and M.B.; software, S.K.; validation, S.K.; formal analysis, S.K.; investigation, S.K.; resources, S.K.; data curation, S.K.; writing—original draft preparation, S.K. and M.B.; writing—review and editing, S.K. and M.B.; visualization, S.K.; supervision, S.K.; project administration, S.K.; funding acquisition, S.K. All authors have read and agreed to the published version of the manuscript.

**Funding:** This research received no external funding.

**Institutional Review Board Statement:** The study was conducted in accordance with the Declaration of Helsinki, and ethical review and approval were waived by the Department of Work and Organizational Psychology at Kiel University.

**Informed Consent Statement:** Informed consent was obtained from all subjects involved in the study.

**Data Availability Statement:** The original data and analysis script presented in the study are openly available in OSF at https://osf.io/q2xgj/?view_only=6dcd0acfd5004fb2a9a0113a723eda17 accessed on 25 August 2025.

**Conflicts of Interest:** The authors declare no conflicts of interest.

## Abbreviations

The following abbreviations are used in this manuscript:

DBR    Design-Based Research
HRD    Human Resource Development
OBE    Open-book exam
PjBL    Project-Based Learning

## Appendix A

**Table A1.** Student feedback on expectations and needs assessed prior to the course on HRD.

| Variable | Category | Anchor Examples |
|---|---|---|
| Expectations | Practical insights and application | "I would wish that we work in a very practice-oriented way; 'learning by doing', so to speak." "Lots of practical exercises" "Besides all the theory, practice should not be forgotten, or connections to practice should be made." "How things actually work in HRD and what is truly paid attention to there." |
| | Theoretical foundation | "I want to leave the course with a good overview of the HRD process." "Explanation of what HRD is and its relevance..." "That we can connect to content from the past semester and link knowledge with it..." |
| | Career orientation and industry connection | "I wish for insights into what it's like to work in work and organizational psychology." "I would also find it nice if you could share your personal experiences that you have gathered in this area." "...overview of what directions there are..." |
| | Interactive and engaging learning environment | "...good balance between content input and interactive parts..." "Furthermore, a relaxed, curious atmosphere makes a good course for me; participants are involved and encouraged to ask questions." "The course should be lively, many interactions." |
| | Transparency, clear structure and expectations | "clearly defined expectations for the performance to be delivered by the students" "transparent evaluation or grading criteria..." "The workload should be appropriate..." "which topics are relevant for the exam..." |
| Needs | Deepened and up-to-date theoretical knowledge | "Currently, I still lack a lot of knowledge, which I am sure I will gain in the course." "Significantly more specialized knowledge, as the last course was only about personnel selection..." "A repetition and continuation of the theory behind it, and to practice this in practice." |
| | Methodological and practical competence | "Guidelines with the most important steps for HRD" "A more detailed presentation of the methods, context, etc." "Knowledge about the techniques and methods applied in HRD" "...how to implement it." |
| | Room for practicing | "To become active myself and practice in a practical way. So far, it has mostly just been theories..." "Pretty much everything, [...] but otherwise, I particularly lack practical experience." "More practical relevance, concrete case examples on which one can conceptualize, apply, and practice the measures." "To carry out previously learned theory using an example." |

**Table A2.** Student feedback on strengths and limitations of the course, and needs assessed after the course on HRD.

| Variable | Category | Anchor Examples |
|---|---|---|
| Strengths of the course | Practical orientation and examples | "Repeated practical relevance and clarification using examples (relevance became clearer)." |
| | Project was not graded | "I didn't think that not being graded would take so much pressure off the project work." |
| | No (graded) oral presentation | "You had a real task to do, not just give a presentation, which is often the case." |
| | Open-book exam | "In the end, I thought the exam was very well designed, and it was better that it was graded rather than the intervention or the report." |
| | Structure or concept of the course | "You are not overwhelmed with information; instead, each session feels like you are being given relevant content in manageable portions." |
| | Interaction and teamwork | "interactive concept"<br>"teamwork" |
| | Interesting and very varied course | "interesting"<br>"very varied" |
| Limitations of the course | Heavy workload | "In addition, the effort required for this course (I think it was only worth 4 credits?) was unreasonably high. Two preliminary assignments and a final exam? At least one of these could and should have been removed. Especially in the last semester, when some students are writing their bachelor's thesis and trying to get the best out of their final exams in order to have a chance at the master's degree they are aiming for. If this had been a course in the first semester, I would have evaluated it differently. Or if there were more credits. But as it was, it was simply disproportionate and not student-friendly." |
| | Unclear communication regarding expectations and requirements | "Unclear and changing communication regarding the exam; the requirements for depth/detail in the report were not always entirely clear." |
| | Structure of the course | "Learning objectives before each session to mentally narrow down the input" |
| | Short time between content delivery and application | "Too little time to directly and creatively implement what was learned into the project work ('creativity on demand')." |
| | Excessive proportion of lecturer given by the educator | "A lot of lecturing by the educator." |
| | Feedback culture | "Feedback culture" |
| Needs | More hands-on experience | "Probably just more experience, so not necessarily something the course could have changed." |
| | Further courses on HRD | "Further (project) work/application with practical relevance and more than 1–2 courses on this topic"<br>"More practical application courses." |
| | Feedback on the implementation of the intervention | "Feedback on our HRD intervention, which we conducted at the end (implementation, trainer behavior)." |
| | Gain insight into practice through others | "More practical (realistic) insights, for example, through speeches/examples from experts." |

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
