# Peer review of "Empowering Future HR Professionals: A Design-Based Research Approach to Project-Based Learning in Work and Organizational Psychology"

_education, doi:10.3390/educsci15101337_

Round 1

Reviewer 1 Report

Comments and Suggestions for Authors

The article presents an innovative and well-structured attempt to integrate project-based learning, open educational practices, and generative AI into a Human Resource Development course.   It is commendable for addressing a highly relevant question in higher education: how to prepare students for complex professional environments by combining authentic project work with open and participatory assessment formats. The study demonstrates strong pedagogical creativity, especially in involving students in the co-creation of exam tasks and in leveraging generative AI for case development and interview simulations. At the same time, several areas could be improved to enhance the clarity and rigor of the work:

  1. The pre-test involved 28 participants and the post-test 23, yet the reasons for the missing data and the reduction to 16 matched cases are not explained.  Greater transparency regarding attrition would strengthen the evaluation.
  2. Although the study is presented as design-based research, the iterative process is not sufficiently visualized.  A flowchart or iteration diagram would make the DBR cycle clearer and illustrate how future redesigns will be approached.
  3. Introducing relevant motivational theories would provide stronger theoretical grounding and help interpret the observed findings.
  4. The text contains multiple instances of “HDR,” which appear to be typographical errors for “HRD.”  These should be corrected for consistency.
  5. While the use of self-report scales is described, there is no information about their reliability or validity.  Reporting these metrics is essential to support the robustness of the conclusions.

Addressing these points would substantially improve the article’s methodological transparency, theoretical grounding, and overall persuasiveness.

Author Response

Comments and Suggestions for Authors

The article presents an innovative and well-structured attempt to integrate project-based learning, open educational practices, and generative AI into a Human Resource Development course.   It is commendable for addressing a highly relevant question in higher education: how to prepare students for complex professional environments by combining authentic project work with open and participatory assessment formats. The study demonstrates strong pedagogical creativity, especially in involving students in the co-creation of exam tasks and in leveraging generative AI for case development and interview simulations. At the same time, several areas could be improved to enhance the clarity and rigor of the work:

The pre-test involved 28 participants and the post-test 23, yet the reasons for the missing data and the reduction to 16 matched cases are not explained.  Greater transparency regarding attrition would strengthen the evaluation.

Response: We thank the reviewer for this valuable comment. The revised manuscript now provides a clearer explanation of participant attrition and the reduction to 16 matched cases. As described in Section 5.1 (Participants and Procedure), survey participation was voluntary and conducted without incentives, which led to 28 responses at pre-test and 23 at post-test. In addition, several students did not record their personal ID during the first survey, which prevented matching their responses across time points. Consequently, only 16 complete matched cases were available for analysis.

Although the study is presented as design-based research, the iterative process is not sufficiently visualized.  A flowchart or iteration diagram would make the DBR cycle clearer and illustrate how future redesigns will be approached.

Response: We thank the reviewer for this helpful suggestion and fully agree that a visual representation strengthens the manuscript. Accordingly, we created a flow chart to illustrate the DBR cycle (Figure 1). In addition, responding to another reviewer’s observation that the theoretical background is dense and sometimes difficult to navigate, we also developed an illustration of the PjBL characteristics embodied in the course design (Figure 2). These figures have been added to the manuscript and are intended to improve clarity and enhance readability by complementing the text with visual summaries. In addition, as suggested, we have also reviewed and improved the quality of our English.

Introducing relevant motivational theories would provide stronger theoretical grounding and help interpret the observed findings.

Response: We fully agree with this valuable suggestion. In response, we have expanded the new Section 2 (Open pedagogy as a framework for educational innovation) by integrating relevant motivational theories. Specifically, we discuss self-determination theory, expectancy-value theory, achievement goal theory, and flow theory to show how principles of open pedagogy—such as learner agency, collaboration, and authenticity—support intrinsic motivation and engagement. This addition provides a stronger theoretical grounding and helps to interpret the observed findings considering established psychological frameworks.

The text contains multiple instances of “HDR,” which appear to be typographical errors for “HRD.”  These should be corrected for consistency.

Response: Thanks for capturing this. Consequently, we corrected all instances.

While the use of self-report scales is described, there is no information about their reliability or validity.  Reporting these metrics is essential to support the robustness of the conclusions.

Response: We appreciate this important point. In Section 5.2.1.1 (Quantitative Data) and Section 7 (Limitations), we now explicitly clarify that our items were self-developed to directly reflect the course’s learning objectives and intended professional qualifications or were adapted from institutional evaluation, and were therefore designed for descriptive monitoring rather than psychometric validation. As such, we do not report internal consistency coefficients, because the items do not represent multi-item scales. Instead, we emphasize that findings should be interpreted as exploratory indicators of change. To strengthen transparency, we also now discuss this limitation explicitly, noting the absence of standardized psychometric validation and encouraging future research to include validated instruments and behavioral assessments.

Addressing these points would substantially improve the article’s methodological transparency, theoretical grounding, and overall persuasiveness.

Reviewer 2 Report

Comments and Suggestions for Authors

The chosen research problem is relevant not only for educational practice, but also for theory, because another educational strategy is theoretically and empirically substantiated. The topic, course and results of the research are presented, analyzed in detail and properly. The article has an appropriate structure, the research sample is presented in detail, the methods and the research course are named. This creates reasonable assumptions for the objectivity of the research.
Perhaps the research results could be presented in a more statistically informative way.
The conclusions and discussion are broad and detailed, but there is a lack of recommendations that would be especially relevant for educational practitioners.

Author Response

Comments and Suggestions for Authors

The chosen research problem is relevant not only for educational practice, but also for theory, because another educational strategy is theoretically and empirically substantiated. The topic, course and results of the research are presented, analyzed in detail and properly. The article has an appropriate structure, the research sample is presented in detail, the methods and the research course are named. This creates reasonable assumptions for the objectivity of the research.
Perhaps the research results could be presented in a more statistically informative way.
The conclusions and discussion are broad and detailed, but there is a lack of recommendations that would be especially relevant for educational practitioners.

Response: We thank the reviewer for these constructive suggestions.

First, regarding the presentation of results, we have revised the Results section (5.2.1.1 Quantitative Data and Table 3) to make the statistical information more transparent and informative. Specifically, we now report Wilcoxon signed rank test outcomes with effect sizes (r) to provide a clearer picture of the magnitude of change across pre- and post-measures. More importantly, we have added subheadings (distinguishing between quantitative and qualitative results) and moved some parts of the text to the corresponding subheadings. This strengthens the statistical informativeness of the findings.

Second, we agree that explicit recommendations for practitioners would add value. While many practical insights were already embedded in our Reflection (5.3) and Discussion (6) sections, we have now synthesized these points in a dedicated Section 8 (Practical implications for educational practitioners). This section highlights key takeaways for educators implementing PjBL with open pedagogy, such as the importance of clear communication of requirements, supporting intrinsic motivation, embedding formative feedback cycles, and considering the role of AI for authentic task design. By presenting these as explicit implications from our empirical findings and reflections, we aim to provide actionable guidance for educational practice.

Reviewer 3 Report

Comments and Suggestions for Authors

This manuscript presents a timely and thoughtful exploration of educational innovation, with a strong theoretical foundation and clear empirical engagement. The integration of open pedagogy and project-based learning within the Human Resource Development (HRD) context is relevant and promising. Several aspects of the work could be strengthened to improve clarity, coherence, and academic contribution.

  1. The paper refers to key concepts of open pedagogy, such as co-creation, transparency, and participation. However, the relationship between open pedagogy and project-based learning is not fully developed. It would be helpful to clarify how these two frameworks relate to each other, especially in terms of their practical application in course design. The literature review includes many valuable references, but the narrative feels somewhat scattered. A more cohesive synthesis could help organize the theoretical foundation and highlight the manuscript’s unique contribution.
  2. The methodology section would benefit from greater depth. The sample size is relatively small, with 31 participants, all of whom are female. This demographic limitation may affect the generalizability of the findings and should be addressed more explicitly in the limitations. The use of the Wilcoxon signed-rank test is appropriate for measuring pre-post differences, but the manuscript does not report effect sizes or consider potential confounding variables such as prior experience or learner motivation. On the qualitative side, student feedback is presented descriptively without evidence of systematic coding or thematic analysis. Including illustrative quotes and clarifying the analytic process would improve the transparency and credibility of the findings.
  3. The manuscript emphasizes real-world and practice-oriented learning, which is commendable. However, the HRD intervention appears to be limited to a 45-minute session. This raises questions about the depth of workplace engagement and how authenticity is defined and enacted in this context. It may be useful to elaborate on what authenticity means in practice and how it was operationalized in the study. The use of artificial intelligence in case generation and assessment design is a promising feature. Further discussion of its potential in personalized feedback or learning analytics would be welcome.
  4. Some sections of the manuscript, especially in Chapters Three and Four, are quite dense and difficult to navigate. Breaking up longer paragraphs and incorporating visual elements could improve readability. Terminology such as "open pedagogy," "participatory learning," and "co-creation" is used throughout the text, but not always consistently. A clearer and more uniform use of these terms would help maintain conceptual coherence.
  5. The reflection section offers useful insights but tends to be descriptive rather than critical. It would be helpful to engage more deeply with the tensions between educational innovation and institutional constraints or to situate the reflection within broader theoretical conversations.

This is a promising manuscript with strong potential. The suggestions above are offered with the hope that they will support the author in further refining the work.

Author Response

Comments and Suggestions for Authors

This manuscript presents a timely and thoughtful exploration of educational innovation, with a strong theoretical foundation and clear empirical engagement. The integration of open pedagogy and project-based learning within the Human Resource Development (HRD) context is relevant and promising. Several aspects of the work could be strengthened to improve clarity, coherence, and academic contribution.

The paper refers to key concepts of open pedagogy, such as co-creation, transparency, and participation. However, the relationship between open pedagogy and project-based learning is not fully developed. It would be helpful to clarify how these two frameworks relate to each other, especially in terms of their practical application in course design. The literature review includes many valuable references, but the narrative feels somewhat scattered. A more cohesive synthesis could help organize the theoretical foundation and highlight the manuscript’s unique contribution.

Response: We agree that this was a significant shortcoming in the previous version. Consequently, we have made several revisions to strengthen conceptual clarity. First, we added a new section after the introduction (2. Open pedagogy as a framework for educational innovation) that introduces the framework of open pedagogy and related concepts, supported by Table 1. In this section, we also developed the relationship between open pedagogy and PjBL to establish a stronger theoretical foundation.

Second, to provide a clear synthesis, we added an integrating section (4.5. From principles to practice: Synthesizing open pedagogy and PjBL) that explicitly demonstrates how open pedagogy provides the pedagogical orientation (e.g., co-creation, transparency, learner agency, collaboration, inclusivity, formative learning cycles, authenticity), while PjBL functions as the methodological enactment of these principles.

Finally, throughout the manuscript we revised terminology and usage of open pedagogy concepts to consistently match the definitions provided in Table 1. These changes ensure that the manuscript now presents a coherent and well-structured integration of both frameworks, directly addressing your concern.

The methodology section would benefit from greater depth. The sample size is relatively small, with 31 participants, all of whom are female. This demographic limitation may affect the generalizability of the findings and should be addressed more explicitly in the limitations. The use of the Wilcoxon signed-rank test is appropriate for measuring pre-post differences, but the manuscript does not report effect sizes or consider potential confounding variables such as prior experience or learner motivation. On the qualitative side, student feedback is presented descriptively without evidence of systematic coding or thematic analysis. Including illustrative quotes and clarifying the analytic process would improve the transparency and credibility of the findings.

Response: We thank the reviewer for this valuable comment. We fully agree that the methodological section requires greater depth and that the limitations of our approach need to be more explicitly acknowledged. In our revised manuscript, we have:

- Calculated and reported effect sizes for the Wilcoxon signed rank tests to provide additional information about the magnitude of pre–post differences.

- Explicitly discussed the small, predominantly female sample and its implications for generalizability. We also clarified that the survey scales were self-developed or developed by our university for internal evaluation purposes, are single-item measures, and should therefore be interpreted descriptively rather than psychometrically.

- Noted that, given the need to keep the survey brief and the exploratory nature of the study, we did not collect data on prior experience or learner motivation. We acknowledge this as a limitation and suggest it as an avenue for future research.

- Expanded the description of how we analyzed qualitative student feedback. While we did not conduct a full thematic analysis, we categorized open-text responses into recurring themes, and reported frequencies. We included illustrative quotes also in the main text (in addition to the Appendix) to increase transparency and credibility.

- Emphasized that this study represents a preliminary and exploratory case study within a Design-Based Research (DBR) cycle, with the main goal of generating insights for iterative course improvement rather than producing generalizable statistical claims.

We hope that our changes meet your expectations.

The manuscript emphasizes real-world and practice-oriented learning, which is commendable. However, the HRD intervention appears to be limited to a 45-minute session. This raises questions about the depth of workplace engagement and how authenticity is defined and enacted in this context. It may be useful to elaborate on what authenticity means in practice and how it was operationalized in the study. The use of artificial intelligence in case generation and assessment design is a promising feature. Further discussion of its potential in personalized feedback or learning analytics would be welcome.

Response: We thank the reviewer for this insightful comment. We agree that authenticity is a central dimension of PjBL and requires further clarification. In our revision, we have elaborated on how authenticity was conceptualized and enacted in this course (see Section 4.4.2 Authenticity). While each intervention session was limited to 45 minutes due to curricular constraints, students engaged in the full HRD cycle (needs analysis, conception, design, implementation, and evaluation) over the entire semester. Thus, authenticity was operationalized less through the duration of the intervention and more through the processes, roles, and tasks that mirror professional HRD practice. We also explicitly acknowledge this time constraint in the Limitations section, clarifying that authenticity in our study should be understood primarily in a process-oriented sense, emphasizing meaningful engagement with realistic professional tasks and an authentic audience rather than intervention length alone.

We have also expanded the discussion of AI to include its potential for personalized feedback and learning analytics, while acknowledging that this was beyond the scope of the present pilot but represents a promising direction for future work.

Some sections of the manuscript, especially in Chapters Three and Four, are quite dense and difficult to navigate. Breaking up longer paragraphs and incorporating visual elements could improve readability. Terminology such as "open pedagogy," "participatory learning," and "co-creation" is used throughout the text, but not always consistently. A clearer and more uniform use of these terms would help maintain conceptual coherence.

Response: We thank the reviewer for this valuable recommendation, which was of great assistance. We agree that the manuscript was text-heavy in parts. To improve readability, we have broken up longer paragraphs and incorporated two visual elements. First, we added a figure illustrating the DBR cycle followed in this study (Figure 1), as also suggested by another reviewer. Second, we created a figure visualizing the PjBL characteristics embodied in the course design to guide readers more clearly through Section 4.4 (Figure 2). We believe these changes enhance both clarity and accessibility of the manuscript.

Regarding the second point you raised, we fully agree. Accordingly, we added a section after the Introduction to present the framework of open pedagogy and its related concepts, as well as a synthesizing chapter before the evaluation section (see also our response to your first comment).

The reflection section offers useful insights but tends to be descriptive rather than critical. It would be helpful to engage more deeply with the tensions between educational innovation and institutional constraints or to situate the reflection within broader theoretical conversations.

Response: We thank the reviewer for this constructive observation. We agree that the original reflection section tended to be more descriptive, and we have revised it to engage more critically with the broader implications of our findings. Specifically, we added short analytical comments throughout Section 5.3 that highlight the tensions between educational innovation and institutional constraints, such as the conflict between student autonomy and compulsory participation, the challenge of aligning PjBL with rigid curricular structures, and the difficulty of reconciling DBR with conventional standards of rigor. These additions aim to strengthen the critical depth of the reflection while keeping the section concise and accessible.

This is a promising manuscript with strong potential. The suggestions above are offered with the hope that they will support the author in further refining the work.

Round 2

Reviewer 1 Report

Comments and Suggestions for Authors

Thank you for your thorough revisions. I have very few comments left—thank you.

Author Response

Thank you very much for your feedback!

Reviewer 3 Report

Comments and Suggestions for Authors

The manuscript presents a compelling case for the benefits of open pedagogy and project-based learning. To strengthen the discussion, I encourage the authors to also consider limitations noted in the literature, such as increased cognitive load on students and reduced motivation for independent learning. Including a critical perspective, such as the work by Kirschner, Sweller, and Clark (2006), would help provide a more balanced view.

I commend the authors for their thoughtful revisions. The manuscript is much improved, and I have no further suggestions.

Author Response

Comment 1: 

The manuscript presents a compelling case for the benefits of open pedagogy and project-based learning. To strengthen the discussion, I encourage the authors to also consider limitations noted in the literature, such as increased cognitive load on students and reduced motivation for independent learning. Including a critical perspective, such as the work by Kirschner, Sweller, and Clark (2006), would help provide a more balanced view.

Response: We appreciate this insightful comment and fully agree that incorporating a critical perspective on the potential cognitive and motivational challenges of PjBL would enrich the discussion. To address this, we have added a new paragraph in the Discussion section.

This new passage integrates the theoretical perspective of Kirschner, Sweller, and Clark (2006) on cognitive load and minimal guidance with our own empirical reflections, highlighting the importance of scaffolding, feedback, and transparent communication.

The added paragraph reads as follows:

"At the same time, it is important to acknowledge that the inherent freedom and experimental nature of PjBL, while beneficial for fostering autonomy and creativity, can also challenge students by increasing cognitive load and reducing perceived structure. As Kirschner et al. (2006) emphasized, minimal guidance during instruction may overwhelm learners’ working memory, particularly when they are still novices within a given domain.

Our findings resonate with this perspective: the openness of PjBL may have induced resistance and uncertainty among students, occasionally leading to anxiety or feelings of overload. To mitigate these effects, scaffolding—including gradually fading assistance­—, transparent communication, and continuous feedback are essential. Providing clear milestones, explicit expectations, and timely formative guidance helps manage cognitive demands while preserving the participatory and self-directed qualities of open pedagogy. In this sense, our experience confirms that openness and structure are not mutually exclusive but must be intentionally balanced to reduce cognitive overload and sustain intrinsic motivation for independent learning.

Building on this insight, we emphasize the importance of clear communication and transparent requirements."

This addition explicitly discusses the cognitive-load dimension and the need to balance openness with structured support, thus hopefully providing the critical nuance you have requested.